# BioCG: Constrained Generative Modeling for Biochemical Interaction Prediction

**Amitay Sicherman**
Department of Computer Science
Technion - Israel Institute of Technology
Israel
amitay.s@cs.technion.ac.il

**Kira Radinsky**
Department of Computer Science
Technion - Israel Institute of Technology
Israel
kirar@cs.technion.ac.il

## Abstract

Predicting interactions between biochemical entities is a core challenge in drug discovery and systems biology, often hindered by limited data and poor generalization to unseen entities. Traditional discriminative models frequently underperform in such settings. We propose BioCG (Biochemical Constrained Generation), a novel framework that reformulates interaction prediction as a constrained sequence generation task. BioCG encodes target entities as unique discrete sequences via Iterative Residual Vector Quantization (I-RVQ) and trains a generative model to produce the sequence of an interacting partner given a query entity. A trie-guided constrained decoding mechanism, built from a catalog of valid target sequences, concentrates the model's learning on the critical distinctions between valid biochemical options, ensuring all outputs correspond to an entity within the pre-defined target catalog. An information-weighted training objective further focuses learning on the most critical decision points. BioCG achieves state-of-the-art (SOTA) performance across diverse tasks, Drug-Target Interaction (DTI), Drug-Drug Interaction (DDI), and Enzyme-Reaction Prediction, especially in data-scarce and cold-start conditions. On the BioSNAP DTI benchmark, for example, BioCG attains an AUC of 89.31% on unseen proteins, representing a 14.3 percentage point gain over prior SOTA. By directly generating interacting partners from a known biochemical space, BioCG provides a robust and data-efficient solution for in-silico biochemical discovery.

## 1 Introduction

Predicting whether two biochemical entities interact—for example, whether a drug will affect its protein target or if an enzyme will catalyze a given reaction—is pivotal for drug discovery, synthetic biology, and basic biochemistry [18, 36, 38]. Accurate in-silico models dramatically cut experimental cost and time while guiding the rational design of new therapeutics and pathways [2, 30, 4]. Most machine-learning systems cast interaction prediction as binary classification: they embed each entity independently and train a discriminative head to score compatibility [21, 41, 33, 20]. While this paradigm performs well with abundant labeled data, it breaks down in the low-data regimes common in biochemical research. Interaction labels are often scarce due to the high cost of experimental validation [34, 9]. Moreover, discriminative models typically fail to generalize in cold-start scenarios, where the interacting entities have not been seen during training [6, 23]. Generative models are often seen as a promising alternative, capable of achieving strong results by learning complex underlying data distributions. However, this typically requires large training corpora, and a further significant challenge is that unconstrained generative approaches tend to produce outputs that violate structural or chemical rules [8, 19]. Biochemical interaction prediction occupies a unique position where potential interaction partners are drawn from a finite catalog of valid entities. For instance, predicting

39th Conference on Neural Information Processing Systems (NeurIPS 2025).

an enzyme's function for a novel reaction often involves selecting a known enzyme, which is typically cataloged by its sequence and classified by an Enzyme Commission (EC) number, from a finite set such as the human proteome [14, 31]. Crucially, these catalog entries are not simple labels but rich, structured objects (such as protein sequences or molecular graphs) that can be represented as token sequences,[26, 39]. Therefore, we propose reframing biochemical interaction prediction as constrained sequence generation over a finite set of valid, structured entities.

We present BIOCG that exploits this insight by representing every catalog entry as a compact code sequence obtained via I-RVQ. A trie built over these codes guides the generative model: during training, it directs learning to focus on distinguishing between legitimate candidates, and during inference, it ensures only sequences corresponding to entities from the catalog are emitted. A lightweight meta-calibration model then processes features derived from the generative model's process to output final interaction scores.

The contributions of this work are threefold. First, we introduce BIOCG, a novel framework reformulating biochemical interaction prediction as constrained sequence generation over a finite catalog of valid entities. Second, we develop its core technical innovations: (i) Iterative Residual Vector Quantization (I-RVQ) to create unique discrete codes for target entities; (ii) a trie-guided constrained decoding mechanism, active during training and inference; and (iii) an information-weighted training objective that enhances learning efficiency and generalization. Third, we demonstrate state-of-the-art performance across key biochemical benchmarks (including DTI, DDI, and Enzyme-Reaction Prediction), with up to 14 percentage points AUC improvement in challenging scenarios. To facilitate full reproducibility of all experiments, we release our open-source code (GitHub).

## 2    Related Work

**Drug-Target Interaction Prediction.** Predicting drug-target interactions (DTI) is crucial for drug discovery. Deep learning has significantly advanced DTI prediction using diverse architectures, including Convolutional Neural Networks (CNNs) [16], Graph Neural Networks (GNNs) for molecular topology [22], Transformers for biochemical sequences [13], and various multimodal or attention-based approaches [1, 17, 33, 20]. These methods frame DTI as a discriminative task. BioCG differs by employing constrained sequence generation over known targets, improving cold-start generalization.

**Drug-Drug Interaction Prediction.** Predicting drug-drug interactions (DDIs) is crucial for patient safety. Machine learning models identify DDIs using drug similarities [29] or Graph Neural Networks (GNNs) on molecular structures [42, 37]. These typically classify pre-defined drug pairs. BioCG's framework conceptually differs by generating an interacting drug partner from a known set.

**Enzyme-Reaction Prediction** Identifying the specific enzyme responsible for catalyzing a given biochemical reaction (often classified by its Enzyme Commission (EC) number) is vital for biochemical annotation and pathway understanding. The CARE benchmark [41] provides a standard evaluation setting for this task. Existing methods often use contrastive learning [41, 21] or large language models [25, 41] to align reaction and enzyme representations, typically followed by classification or similarity-based identification of the most likely enzyme/EC number. BioCG offers a distinct approach by directly generating the catalytic enzyme, conditioned on the input reaction.

**Constrained Generation.** Constrained sequence generation aims for valid outputs. Many approaches enforce constraints only during inference [11, 3, 12] or require large datasets to learn constraints implicitly during training [43, 7]. BioCG distinctively applies explicit trie constraints from finite biochemical catalogs during training and inference. This enables data-efficient learning by focusing training on the known valid space and guarantees that outputs correspond to valid entities at inference.

## 3    Methods

We introduce BioCG (Biochemical Constrained Generation), a novel framework designed to predict biochemical entity-entity interactions by reformulating the task as a constrained, autoregressive sequence generation problem. The overall framework is depicted in Figure 1. It involves representing query entities using pre-trained encoders, transforming target entities into unique discrete sequences via I-RVQ, generating these sequences using a trie-guided constrained decoder, and converting sequence generation probabilities into interaction scores.

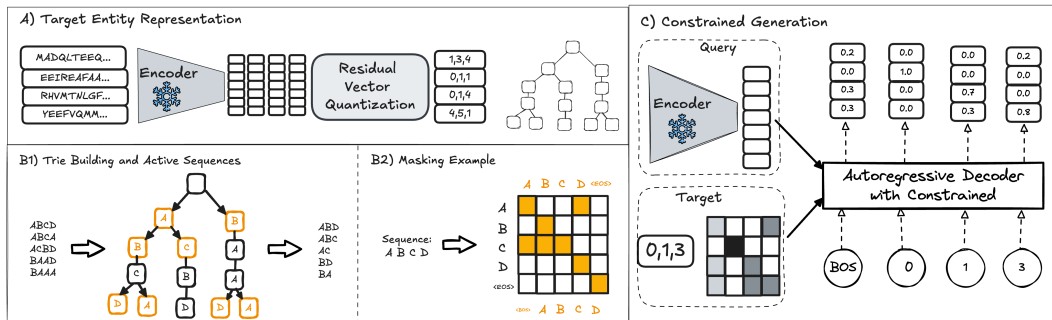

Figure 1: The BioCG framework. (A) Target entity representation pipeline: Target entities (e.g., proteins) are encoded using a pre-trained model. I-RVQ iteratively clusters residuals to create unique discrete sequences for each target. (B1) Trie Construction: A prefix tree (trie) is built from all discretized target sequences. Nodes with multiple children (orange) represent active decision points where the model must predict the next token among valid options. (B2) Constrained Decoding Mask: The trie determines valid next tokens at each generation step. An attention mask allows the decoder only to consider valid continuations (orange cells), effectively pruning invalid generation paths. (C) Constrained Generation Process: A query entity (e.g., a drug) is encoded. The autoregressive decoder uses this context and previously generated tokens to predict the next token in the target sequence, guided by the trie-based mask and incorporating information weighting.

**Problem Formulation** We reformulate biochemical interaction prediction as a constrained sequence generation task. Let $\mathcal{E}_{in}$ be the set of input query entities and $\mathcal{T} = \{t_1, t_2, \ldots, t_N\}$ be a finite, known catalog of valid target entities. Given a query $e \in \mathcal{E}_{in}$, our objective is to identify the interacting target $t \in \mathcal{T}$. BioCG achieves this by learning to generate a unique discrete sequence representation $s_t = (w_1, \ldots, w_M)$ corresponding to the target entity $t$. The core of our model is to estimate the conditional probability $P(s_t \mid e)$ of generating this sequence given the query $e$. Crucially, this generation is constrained such that $s_t$ must always represent a valid entity within the predefined catalog $\mathcal{T}$. This is achieved by modeling the standard autoregressive probability $P(s_t \mid e) = \prod_{k=1}^{M} P(w_k \mid e, w_1, \ldots, w_{k-1})$, while ensuring that only sequences $s_t$ corresponding to an entity in $\mathcal{T}$ can be generated. This constrained sequence generation probability derives the final interaction likelihood between $e$ and $t$.

## 3.1 Model Architecture and Representation

**Query Entity Representation.** Input query entities are represented using standard biochemical sequence formats: proteins via FASTA sequences and small molecules or reactions via SMILES strings. We leverage SOTA pre-trained sequence-based encoders that transform the input query $e$ into a fixed-size context vector $h_{in}$. These encoders are kept frozen during BioCG training to preserve their learned domain knowledge and prevent catastrophic forgetting. Specifically, we utilize ESM [10] for proteins, Molformer [28] for molecules, and RXNFP [32] for reactions. The impact of alternative pre-training strategies and models is explored in ablation studies (Section 6).

**Target Entity Representation via I-RVQ.** While target entities like proteins and molecules possess inherent discrete sequence representations, BioCG employs a strategy to generate potentially more effective sequences tailored for the decoder. This involves first obtaining continuous embeddings for each target entity $t \in \mathcal{T}$ using an appropriate pre-trained encoder. Subsequently, we introduce and utilize I-RVQ, our adaptation of RVQ specifically designed to generate unique discrete code sequences from these embeddings for the entire target set. Our I-RVQ approach leverages the standard mechanism of RVQ: it iteratively quantizes residual vectors using k-means clustering ($K$ clusters per layer). In each layer $m$, an entity's residual embedding $e_i^{(m-1)}$ is assigned to the nearest centroid $c_{i_k^{(m)}}^{(m)}$, yielding the code $i_k^{(m)}$ for that layer, and the new residual $e_i^{(m)}$ is calculated by subtracting the assigned centroid vector. The key adaptation defining I-RVQ lies not in the per-layer quantization step but in its overall iterative application and objective. We apply these I-RVQ layers sequentially ($M = 1, 2, \ldots$) to achieve representation uniqueness across the entire finite target set $\mathcal{T}$. The process continues, adding layers and thus lengthening the code sequences $(i_k^{(1)}, \ldots, i_k^{(M)})$, precisely until every target entity $t_i \in \mathcal{T}$ possesses a distinct code sequence. (See example in Figure 2)

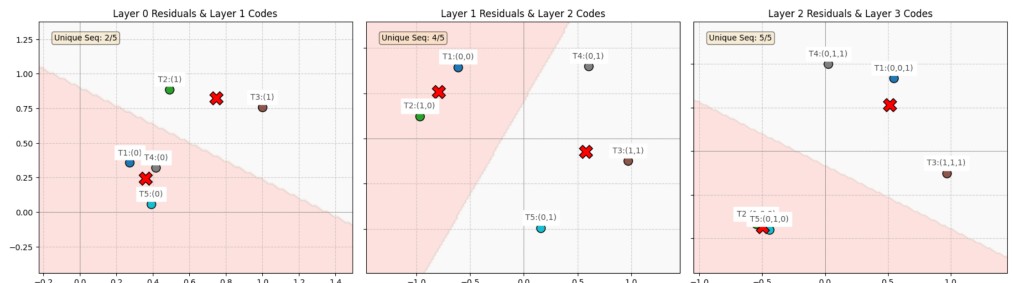

Figure 2: Simulation of I-RVQ with five target entities, using two k-means clusters per layer. Each panel depicts a k-means clustering step—where shaded areas represent cluster boundaries and red X's mark centroids—applied to the residual vectors from the previous layer. Labels indicate each target's ID and the progressively forming code sequence. The iterative process culminates in unique codes for all five targets.

The motivation for generating these learned I-RVQ codes instead of raw sequences is that they can capture salient features in a structured, often more compact format suitable for the Transformer decoder, potentially using a more uniform vocabulary and sequence length. This uniqueness-driven I-RVQ process guarantees the bijective mapping essential for our framework. As validated in ablation studies (Section 6), these I-RVQ codes prove more effective for the generative decoder than alternatives such as raw biological sequences, random assignments, or standard sub-word tokenization techniques like Byte Pair Encoding (BPE).

**Decoder Architecture with Trie-Guided Constrained Decoding.** BioCG's generative core, a Transformer decoder [35], autoregressively generates the target's unique code sequence $s_t = (w_1, \ldots, w_M)$ conditioned on the query context $h_{in}$. A key feature is its rigorous output validity, achieved by confining generation (during training and inference) strictly to known valid target sequences in $\mathcal{T}$ (Fig. 1C). This constrained approach involves:

*1. Trie Construction:* A pre-constructed prefix tree (trie) $\mathcal{G}_T$, built from all valid target code sequences in $\mathcal{T}$ (Fig. 1B1), defines the valid output space by encapsulating their vocabulary and structure. *2. Trie-Guided Masking:* At each generation step $k$, the decoder produces raw logits $l_k$ (over I-RVQ vocabulary $\mathcal{V}$) based on $h_{in}$ and preceding tokens $(w_1, \ldots, w_{k-1})$. The trie $\mathcal{G}_T$, queried with this partial sequence, provides a binary mask $m_k \in \{0,1\}^K$, where $m_k[i] = 1$ if token $i$ is a valid continuation (Fig. 1B2). *3. Constraint Application:* This mask $m_k$ is applied to logits $l_k$ (in training and inference) via: $p_k = \text{softmax}(l_k - \lambda(1 - m_k))$ where a large $\lambda$ (e.g., $10^9$) effectively zeros out probabilities of invalid next tokens. Thus, $p_k$ is non-zero only for valid continuations from $\mathcal{T}$, focusing learning gradients onto valid pathways.

*4. Constrained Training Loss:* The training loss uses the constrained probability $p_k$ for the ground-truth token $w_k^*$. This directs gradients exclusively towards valid transitions, training the model to learn $\mathcal{T}$'s structure and make informed choices among legitimate candidates. *5. Information-Weighted Loss:* Standard cross-entropy is suboptimal for steps with varying branching factors ($BF_k$). Steps with $BF_k = 1$ (single valid option) offer no discriminative learning, while multi-option steps risk imbalanced loss aggregation. Our Information-Weighted Loss assigns negligible weight to $BF_k = 1$ steps. For multi-branch steps, it applies a weight $W_{\text{info}}(m_k) = \log(BF_k + 1 + \epsilon)$ (where $\epsilon$ is a small constant, e.g., $10^{-9}$, for numerical stability). This logarithmic scaling by $BF_k$ (number of valid choices) focuses learning on the most informative and challenging decision points. The overall training objective minimizes the total information-weighted negative log-likelihood for a ground-truth target sequence $s_t^* = (w_1^*, \ldots, w_M^*)$ given query $e$:

$$\mathcal{L}(s_t^*|e) = -\sum_{k=1}^{M} W_{\text{info}}(m_k) \log p_k[w_k^*]$$

where $p_k[w_k^*]$ is the constrained probability of the true token $w_k^*$ at step $k$. The efficacy of BioCG's core components, including trie-guided constrained generation and information-weighted loss, is validated in our ablation studies (Section 6).

**Converting Generation Probabilities to Interaction Scores.** Once the BioCG generative model is trained, a method is required to derive an interaction score for a given query entity $e$ and a specific

target entity $t$ (represented by its ground-truth I-RVQ sequence $s_t^*$) for evaluation on benchmark datasets. We first obtain the conditional probability of generating $s_t^*$ given $e$ to achieve this. This is done using a teacher-forcing approach by feeding the query $e$ and the ground-truth target sequence $s_t^*$ through the trained generative model. This process yields the sequence of step-wise probabilities $p_k[w_k^*]$ for each token in $s_t^*$, along with other potential features from the generation process. From this information, we employ two primary methods to compute a final interaction score between $e$ and $t$:

1. *Log Probability Score:* A straightforward score is derived from the average log probability of the ground-truth target sequence $s_t^*$, typically focusing on 'active' generation steps where the trie-guided $BF_k > 1$. This score is calculated as: $S_{\text{logprob}} = (|\{k : BF_k > 1\}|)^{-1} \sum_{k:BF_k>1} \log p_k[w_k^*]$. A higher score suggests a stronger predicted interaction.

2. *Meta-Model Calibration:* A lightweight Transformer-based meta-model is utilized for potentially more nuanced binary classification. This meta-model is trained to take a sequence of features extracted from the teacher-forced generation of $s_t^*$ (the token probability $p_k[w_k^*]$, its rank among valid alternatives, the branching factor $B_k$, and entropy at each active step $k$) and output a single, calibrated interaction probability for the pair $(e, t)$.

## 4 Experiments

### 4.1 Datasets

**Drug-Target Interaction (DTI) Prediction.** For DTI, we use the BioSNAP dataset [44] (derived from DrugBank [14]), featuring 13,741 positive interactions between 4,510 drugs and 2,181 proteins, with randomly sampled negatives. We follow established protocols [33, 13] for three splits. In the *random split*, where entities are generally seen during training, BioCG generates a known partner (e.g., a protein from $\mathcal{T}_{\text{proteins}}$) for a given query (e.g., a drug), with $\mathcal{T}$ comprising all relevant entities from the training interaction data. For the *unseen protein split*, proteins absent from training interactions serve as query entities; BioCG then generates an interacting drug from $\mathcal{T}_{\text{drugs}}$ (all drugs seen in training). Symmetrically, in the *unseen drug split*, new drugs (absent from training interactions) are queries, and BioCG generates an interacting protein from $\mathcal{T}_{\text{proteins}}$ (all proteins seen in training). In these cold-start splits, query entities are novel to the training pairs, while the target catalog $\mathcal{T}$ is built from entities that were part of training interactions.

**Drug-Drug Interaction (DDI) Prediction.** DDI prediction is evaluated using data from DrugBank [14], similar to [42], containing 1.95 million interaction pairs among 2,755 unique drugs. To assess cold-start capabilities, our *cold-start drug query split* involves test pairs where one drug is "unseen" (the query, absent from training interactions) and the other is "seen." BioCG's task is to generate this "seen" partner drug from the target catalog $\mathcal{T}_{\text{drugs}}$, which consists of all unique drugs in the dataset (as they are potentially "seen" partners if not the cold query) when given the unseen query drug.

**Enzyme-Reaction Prediction** The Enzyme-Reaction Prediction task seeks to identify the enzyme catalyzing a given biochemical reaction. We use the CARE benchmark [41] for evaluation, which contains 61,766 reaction-enzyme pairings. Enzymes are often classified by EC numbers, where each EC number can represent a class of enzymes performing the same function; thus, any enzyme in that EC class is a correct match. The dataset features 4,960 unique EC numbers, with enzymes typically represented by their protein sequences. We follow CARE's unseen reaction query split, where novel test reactions are input queries. BioCG's goal is to generate the correct protein sequence (corresponding to an enzyme/EC number) from the target catalog, $\mathcal{T}_{\text{EC\_proteins}}$.

### 4.2 Baselines

To evaluate BioCG's performance rigorously, we compare it against a range of established and state-of-the-art baseline methods specific to each of the three prediction tasks. **Drug-Target Interaction (DTI) Baselines:** Baselines for DTI prediction on the BioSNAP dataset include classic machine learning models (Random Forest, SVM) and prominent deep learning architectures. These deep learning methods encompass Convolutional Neural Networks (CNNs) like DeepConv-DTI [16], Graph Neural Networks (GNNs) such as GraphDTA [22], Transformer-based models like MolTrans [13]. Various attention or multimodal fusion approaches including DrugBAN [1], DLM-DTI [17], Top-DTI [33], and DrugLAMP [20].

**Drug-Drug Interaction (DDI) Baselines:** For binary DDI prediction on DrugBank, baselines include Logistic Regression (LR) and Random Forest (RF) using chemical structural similarity profiles (SSP). Deep learning comparators are DeepDDI [29], which uses SSPs with a deep neural network; MR-GNN [40], a multi-resolution graph neural network; SSI-DDI [24], a knowledge-driven model learning substructure interactions; and DDI-GCN [42], which employs Graph Convolutional Networks on chemical structures.

**Enzyme-Reaction Prediction Baselines:** BioCG is compared against several established methods. These include a Random classification baseline, a DRFP-based reaction similarity baseline [27], an LLM-based approach (GPT3 [25, 41]), and contrastive learning frameworks designed for this task, namely CREEP [41] and CLIPZyme [21].

### 4.3 Implementation Details

Our BioCG framework uses a Transformer architecture. It integrates frozen pre-trained models for entity representation: ESM2-650M [10] for protein sequences, MoLFormer-XL [28] for molecules, and RXNFP [32] for reactions. The I-RVQ module utilizes k-means clustering for target discretization. The main model is trained using the AdamW optimizer with a constant learning rate, employing early stopping based on validation AUC. A 1-layer Transformer meta-model calibrates generation statistics into interaction scores via a sigmoid binary classification head. Hyperparameters were optimized through greedy search with cross-validation to maximize validation AUC; detailed information is available in Appendix B. Training is conducted on an L40 GPU (64GB), with the main model requiring approximately 6 hours and the meta-model around 5 minutes. The complete implementation of BioCG is publicly available at GitHub.

**Computational Complexity** BioCG's computational profile has two phases. *Offline Preprocessing* involves encoding all target entities in $\mathcal{T}$, running I-RVQ, and constructing the prefix trie $\mathcal{G}_T$. This is a one-time, potentially intensive cost. *Online Inference* for a query $e$ involves fast encoding of $e$ and autoregressive generation via the constrained decoder. Trie lookups for masking add negligible overhead to the Transformer's per-step complexity. The main memory consideration is the storage of the trie $\mathcal{G}_T$. Its size scales with the total number of nodes, which depends on $|\mathcal{T}|$ and the sequence length $M$, potentially becoming large for huge target sets. However, for moderately sized biochemical catalogs (e.g., thousands to tens of thousands of proteins or known drugs), the trie often remains manageable relative to the size of large pre-trained encoders or the decoder itself. Addressing scalability to truly massive target sets is an area for future work (Section 6).

### 4.4 Statistical Analysis

Specific statistical tests were employed to determine the statistical significance of performance differences between BioCG and baseline methods tailored to the metric type. The Two-Proportion Z-Test was used for proportion-based metrics (e.g., Accuracy, Sensitivity, Specificity). This test assesses whether the observed success rates of two models (BioCG and a baseline) are significantly different under the null hypothesis that their actual success rates are equal. DeLong's test [5] was used for AUC comparisons. This non-parametric method is designed to compare the AUCs of ROC curves when the models are evaluated on the same test set (leading to correlated results). It tests the null hypothesis that the AUCs of the two models are equal. For all tests, a p-value less than an alpha level of 0.05 was considered to indicate a statistically significant difference. This supports claims of BioCG's superiority, typically assessed using a one-tailed test where the alternative hypothesis is that BioCG's performance is greater than the baseline's.

## 5 Results

The evaluation of BioCG's performance across the different tasks utilizes metrics commonly established and reported in previous works within each domain, ensuring robust and comparable assessment. We report the Area Under the AUC, Sensitivity, and Specificity for Drug-Target Interaction prediction conducted on a balanced dataset. Due to the imbalanced nature of the Drug-Drug Interaction dataset, our evaluation includes the AUPR and Accuracy, in addition to AUC, to provide a comprehensive view of performance. For the Enzyme-Reaction Prediction task, where the goal is often to identify a single correct enzyme for a given reaction, Top-k Accuracy is the primary metric.

Table 1: Drug-Target Interaction Prediction performance on the BioSNAP dataset across seen, unseen protein, and unseen drugs scenarios. Results are reported as AUC, Sensitivity (Sens), and Specificity (Spec). Performance figures in **bold** indicate that the result is the best in its respective column and represents a statistically significant improvement (p < 0.05) over the next best baseline method, with significance determined as detailed in Section 4.4.

| Method | Seen | | | Unseen Protein | | | Unseen Drug | | |
|---|---|---|---|---|---|---|---|---|---|
| | AUC | Sens | Spec | AUC | Sens | Spec | AUC | Sens | Spec |
| Random Forest | 86.01 | 82.37 | 78.63 | 68.71 | 65.82 | 62.93 | 83.51 | 79.87 | 76.32 |
| SVM | 86.21 | 71.13 | 84.17 | 68.83 | 56.91 | 67.34 | 83.74 | 69.07 | 81.64 |
| DeepConv-DTI | 88.61 | 76.07 | 85.14 | 69.21 | 60.81 | 68.14 | 85.61 | 73.72 | 82.54 |
| GraphDTA | 88.73 | 74.51 | 85.42 | 70.43 | 59.62 | 68.32 | 85.82 | 72.31 | 82.87 |
| MolTrans | 89.52 | 81.83 | 83.19 | 71.42 | 65.43 | 66.52 | 85.63 | 79.46 | 80.63 |
| DrugBAN | 90.31 | 82.03 | 84.72 | 71.03 | 65.67 | 67.83 | 88.61 | 80.47 | 83.01 |
| DLM-DTI | 91.42 | 84.87 | 84.43 | 73.04 | 67.82 | 67.53 | 88.73 | 82.37 | 81.93 |
| Top-DTI | 93.91 | 86.63 | 85.73 | 75.01 | 69.31 | 68.67 | 91.25 | 84.03 | 83.17 |
| DrugLAMP | 91.73 | 84.43 | 85.57 | 73.23 | 67.54 | 68.42 | 89.06 | 81.92 | 82.93 |
| BioCG (Our) | **96.61** | **90.27** | **92.43** | **89.31** | **72.92** | **93.34** | 91.21 | 77.83 | **89.71** |

**Drug-Target Interaction Prediction**    As shown in Table 1, BioCG consistently outperforms all baseline methods across all three BioSNAP dataset splits. In the standard "seen" scenario, BioCG achieves an AUC of 96.61%, representing a statistically significant improvement of 2.7% over the previous state-of-the-art method, Top-DTI (93.91%). This improvement is also reflected in the sensitivity (90.27%) and specificity (92.43%) metrics. The most striking results are observed in the "unseen" scenarios. For unseen protein, where the model must predict interactions for entirely unseen protein targets, BioCG achieves an AUC of 89.31%, a substantial 14.3% increase compared to the best baseline (Top-DTI, 75.01%). This highlights BioCG's strong generalization capability to novel protein entities. Similarly, in the unseen drug scenario, BioCG reaches an AUC of 91.21%, surpassing most baselines, though Top-DTI also performs well here. The performance gains in unseen settings underscore the advantage of BioCG's constrained generative approach, which leverages the structured nature of the target space to make predictions for unseen entities more effectively than discriminative models reliant on embedding similarity.

**Drug-Drug Interaction Prediction**    In predicting Drug-Drug Interactions on the DrugBank dataset, BioCG demonstrates superior performance across all evaluated metrics compared to established baseline methods. The results are detailed in Table 2. BioCG achieves an Accuracy of 77.23%, which is a notable improvement over the best-performing baseline in this metric, DDI-GCN (74.05%). This indicates a higher overall correctness in classifying drug pairs as interacting or non-interacting. For the AUC metric, BioCG scores 84.12%, significantly surpassing the highest baseline score of 80.91% achieved by MR-GNN. This highlights BioCG's enhanced ability to discriminate between positive and negative interaction pairs across various thresholds. Furthermore, BioCG obtains an AUPR of 72.16%, which is also higher than the best baseline AUPR of 69.47% (from MR-GNN), underscoring its robustness in scenarios with potentially imbalanced class distributions. These consistent improvements across ACC, AUC, and AUPR suggest that BioCG's constrained generative approach, modeling one drug as a query and generating the potential interacting partner from a known set, effectively captures complex interaction patterns for DDI prediction.

**Enzyme-Reaction Prediction**    For the task of Enzyme-Reaction Prediction from biochemical reactions, evaluated on the CARE benchmark, BioCG demonstrates strong performance, particularly in accurately identifying the correct enzyme within the top few predictions. The results are presented in Table 3.

BioCG achieves a Top@1 accuracy of 67.93%, significantly outperforming the next best baseline, CREEP, which scored 60.32%. This represents a substantial improvement in correctly identifying the exact enzyme as the top prediction. The advantage of BioCG extends across other Top-k metrics. For instance, BioCG achieves a Top@2 accuracy of 82.32% and a Top@3 accuracy of 89.21%, surpassing CREEP (79.46% Top@2, 87.58% Top@3). While CREEP shows strong performance at higher k values, BioCG maintains a competitive edge or comparable performance, for example,

Table 2: Drug-Drug Interaction Prediction performance on the DrugBank dataset. Results are reported as Accuracy (ACC), AUC, and AUPR. Performance figures in **bold** indicate that the result is the best in its respective column and represents a statistically significant improvement ($p < 0.05$) over the next best baseline method, with significance determined as detailed in Section 4.4.

| Method | ACC | AUC | AUPR |
|---|---|---|---|
| LR | 51.73 | 64.57 | 50.69 |
| RF | 60.78 | 73.42 | 60.15 |
| DeepDDI [29] | 72.24 | 80.86 | 69.03 |
| MR-GNN [40] | 71.69 | 80.91 | 69.47 |
| SSI-DDI [24] | 71.08 | 75.34 | 68.62 |
| DDI-GCN [42] | 74.05 | 79.82 | 69.08 |
| BioCG (Ours) | **77.23** | **84.12** | **72.16** |

achieving 94.22% at Top@5 and 98.52% at Top@10. By Top@50, both BioCG and CREEP approach near-perfect prediction accuracy (99.32% and 99.15%, respectively). These results highlight BioCG's capability to effectively learn the mapping from reactions to their corresponding enzymes by generating unique protein representations associated with EC numbers, leading to improved prediction accuracy, especially at the critical top ranks.

Table 3: Enzyme-Reaction Prediction performance on the CARE benchmark dataset, measured by Top-k Accuracy. Performance figures in **bold** indicate that the result is the best in its respective column and represents a statistically significant improvement ($p < 0.05$) over the next best baseline method. Top@k values for k>1 are not reported for the LLM and Random baselines, as these methods are designed to produce a single best prediction, making a ranked-list comparison methodologically distinct.

| Method | Top@1 | Top@2 | Top@3 | Top@4 | Top@5 | Top@10 | Top@50 |
|---|---|---|---|---|---|---|---|
| Random [41] | 0.00 | | | | | | |
| DRFP Similarity [27] | 59.39 | 70.01 | 78.46 | 80.92 | 81.98 | 85.56 | 93.42 |
| CREEP [41] | 60.32 | 79.46 | 87.58 | 90.39 | 92.10 | 96.72 | 99.15 |
| LLM [25] | 13.74 | | | | | | |
| CLIPZyme [21] | 12.27 | 21.43 | 26.73 | 32.31 | 35.96 | 53.70 | 78.40 |
| BioCG(Our) | **67.93** | **82.32** | **89.21** | 90.52 | **94.22** | **98.52** | 99.32 |

## 6 Ablations

To validate the contributions of BioCG's key components and design choices, we conducted comprehensive ablation studies. These systematically evaluated the impact of different pre-trained encoders and their training strategies, various methods for target entity representation, and alternative constrained decoding approaches. All ablation experiments were performed on the BioSNAP dataset's "unseen protein" split. The findings are summarized in Table 4.

Table 4: Ablation Study Results on BioSNAP Unseen Protein Dataset (AUC). Bold indicates the optimal configuration or best performing model in a category for BioCG.

| Pre-training Model | Pre-training Strategy | Target Representation | Constraint Strategy |
|---|---|---|---|
| **ESM (protein) (89.31)** | Fine-tuned (81.17) | Raw (82.06) | Unconstrained (75.21) |
| ProteinBERT (protein) (88.43) | Scratch (86.26) | Random (79.32) | Inference-only (77.43) |
| GearNet (protein) (87.54) | **Frozen(BioCG) (89.31)** | BPE (84.75) | Cross-Entropy Loss (86.13) |
| **MolFormer (molecule) (89.31)** | ———————— | **I-RVQ-15 (89.31)** | **Full (BioCG) (89.31)** |
| ChemBERTa (molecule) (88.29) | Log Prob BioCG (87.73) | I-RVQ-25 (88.03) | |

**Query Entity Encoders and Training Strategy:** The encoder integration strategy critically affected performance. Optimal AUC (up to 89.31%, Table 4) was achieved using *frozen* high-quality pre-trained encoders (e.g., ESM, MolFormer), which supply rich, stable features. Conversely, *fine-tuning* encoders substantially degraded performance (81.17% AUC). Significantly, training encoders from *scratch* still produced a respectable 86.26% AUC. This indicates that while pre-trained models are beneficial for top results, BioCG's core framework is robust, performing strongly even without them—an advantage if suitable pre-trained models are unavailable. Furthermore, the specific choice among tested pre-trained models (ESM, ProteinBERT, GearNet; MolFormer, ChemBERTa), when used *frozen*, exerted a relatively minor influence on AUC scores (ranging 87.54%-89.31%). This observation reinforces BioCG's architectural robustness to the precise encoder selection, assuming a strong foundational representation.

**Target Entity Representation:** The method for creating discrete sequence representations of target entities for the decoder is crucial. Our I-RVQ with $K = 15$ clusters (**I-RVQ-15**) proved optimal (89.31% AUC). This significantly outperformed using *raw* biological sequences (82.06% AUC when adapted for a comparable baseline) or *random* sequence assignments (79.32% AUC), highlighting the need for structured, learned representations. Further comparisons demonstrated the superiority of I-RVQ-15: it outperformed Byte Pair Encoding (BPE) with 256 tokens (learned from raw sequences like SMILES or FASTA, yielding 84.75% AUC) and other I-RVQ configurations such as I-RVQ with $K = 25$ clusters (I-RVQ-25, which achieved 88.03% AUC).

**Constrained Decoding and Training Strategy:** BioCG's approach of integrating constraints directly into training is vital. The full model with trie-guided decoding and information-weighted loss (**Full (BioCG)**) achieved 89.31% AUC. In contrast, *unconstrained* generation during training (75.21% AUC) or applying constraints *only at inference* (77.43% AUC) performed substantially worse, confirming that the decoder must learn within the valid output space. Using a standard *Cross-Entropy Loss* (86.13% AUC) was less effective than our information-weighted loss, which better handles varying step complexity. Furthermore, the meta-model for calibrating scores (part of "Full (BioCG)") improved results over using direct log probabilities from the generator ("Log Prob BioCG," 87.73% AUC).

## Conclusion

BioCG redefines biochemical interaction prediction as a constrained sequence generation task. By representing target entities as unique discrete sequences via I-RVQ and using a trie-guided constrained decoding mechanism, BioCG ensures valid outputs and focuses learning. An information-weighted training objective further boosts efficiency. Empirical results show BioCG's superiority across diverse tasks like DTI, DDI, and Enzyme-Reaction Prediction, especially in cold-start settings (e.g., a 14.3% AUC rise on BioSNAP DTI for unseen proteins). This highlights the benefit of generating valid partners from a known biochemical space, offering a robust, data-efficient alternative to discriminative models, particularly for novel entities.

**Limitations** BioCG, despite its advantages, has specific limitations. Firstly, a fundamental constraint is the requirement that one side of the interacting pair (the target entity) must be drawn from a pre-defined, closed set. This means our model is not directly applicable to scenarios where both interacting entities are novel, such as predicting interactions between two entirely new drugs. Furthermore, BioCG is designed for binary interaction prediction. Extending the framework to generate richer, quantitative outputs, such as binding affinity or the 3D structure of the binding complex, remains an important direction for future research. Secondly, the optimal performance of BioCG, as demonstrated in our ablation studies, is achieved when utilizing pre-trained embeddings. While the framework can operate with encoders trained from scratch or by using raw sequences for target representation, its effectiveness may be reduced in scientific domains where high-quality, relevant pre-trained models are not readily available.

**Future Work** Future research will focus on extending BioCG's capabilities and scope. One key direction is the exploration of alternative discretization techniques for target entities, beyond I-RVQ, and a thorough investigation of how these different methods affect the model's predictive performance and representational efficiency. Another important avenue is to extend the BioCG framework to other sub-fields within biochemical research and to assess its applicability and potential benefits in entirely

different scientific disciplines, such as physics or materials science. Furthermore, we aim to adapt and expand the framework to accommodate more complex interaction types, for example, by enabling the prediction of interactions involving multiple entities simultaneously.

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

# A    Additional Experiments and Analyses

## A.1    Impact of Encoder Scale on Performance

To further clarify the relationship between encoder quality and model performance, we conducted an additional ablation study using ESM2 models of varying sizes on the BioSNAP unseen protein split. The results, presented in Table 5, demonstrate a clear, positive correlation: BioCG's performance robustly scales with the quality of the feature encoder. Notably, the architectural advantage of BioCG is so significant that even when trained from scratch (see Table 4), it surpasses the previous fully-equipped SOTA. This analysis confirms that while better encoders provide better features, a substantial performance lift comes from BioCG's novel constrained generative formulation.

Table 5: Performance (AUC) on the BioSNAP unseen protein split with ESM2 encoders of varying sizes.

| Encoder | Parameters | AUC (unseen protein) |
|---|---|---|
| ESM2 (used in paper) | 650M | 89.31% |
| ESM2 | 150M | 87.14% |
| ESM2 | 35M | 80.92% |
| ESM2 | 8M | 74.33% |
| Top-DTI (Previous SOTA) | N/A | 75.01% |

## A.2    Quantitative Failure Case Analysis

To better understand the model's failure modes, we performed an analysis of the token index at which the first prediction error occurs during the generation of an incorrect I-RVQ sequence. In the RVQ structure, earlier tokens correspond to coarse-grained clusters representing broad biochemical properties, while later tokens make fine-grained distinctions between highly similar entities. Our analysis reveals that the majority of errors occur late in the sequence:

- First Quarter of sequence: 12% of errors

- Second Quarter of sequence: 18% of errors

- Third Quarter of sequence: 29% of errors

- Final Quarter of sequence: 41% of errors

This distribution strongly suggests that BioCG correctly identifies the general biochemical "neighborhood" of the target and typically only fails at the final, most difficult step of distinguishing between very closely related entities. This indicates the model is making meaningful "close misses" rather than random errors.

## A.3    Conceptual Comparison with Contrastive Learning

While contrastive learning has proven effective for many representation learning tasks, its application to interaction prediction has inherent challenges that BioCG's generative framework is designed to overcome. Contrastive learning aims to learn a general similarity metric by pulling positive pairs closer and pushing negative pairs apart in an embedding space. However, it can struggle with "hard negatives"—entities that are structurally or biochemically very similar to a true interacting partner but do not actually interact. The model may incorrectly learn to place these entities close together, leading to false positives.

BioCG avoids this pitfall by reframing the problem. Instead of learning a global similarity score, it learns a sequential decision process. The trie-guided training forces the model to make explicit choices between valid, and often highly similar, candidates at each step of the generation. The information-weighted loss further sharpens this focus on the most ambiguous and difficult decision points (i.e., where the branching factor is high). This approach directly trains the model to learn the fine-grained, discriminative features necessary to distinguish between "close-miss" candidates, a key reason for its strong performance, especially in cold-start scenarios.

Figure 3: Conceptual illustration comparing a contrastive learning approach with BioCG's generative approach. (Left) Contrastive learning may cluster a "hard negative" close to the positive target in the embedding space. (Right) BioCG learns a decision path to distinguish the correct target from other valid but incorrect options within the trie.

## A.4    Quantitative Analysis of Constrained Decoding Space

We analyze the structural properties of our trie-based generation space. **Active Sequence Length:** For a generation sequence, active positions are those with multiple valid next tokens requiring prediction. The active sequence length is:

$$L_{\text{active}} = \frac{1}{|\mathcal{T}|} \sum_{t \in \mathcal{T}} |\{k : BF_k(t) > 1\}| \tag{1}$$

where $BF_k(t)$ is the branching factor at position $k$ for sequence $t$. **Mean Branching Factor (Active Tokens Only):** The overall branching complexity for active positions is:

$$\bar{BF} = \frac{1}{|\mathcal{T}|} \sum_{t \in \mathcal{T}} \frac{1}{|\{k : BF_k(t) > 1\}|} \sum_{k : BF_k(t) > 1} B_k(t) \tag{2}$$

where the sum is taken only over positions with multiple valid choices.

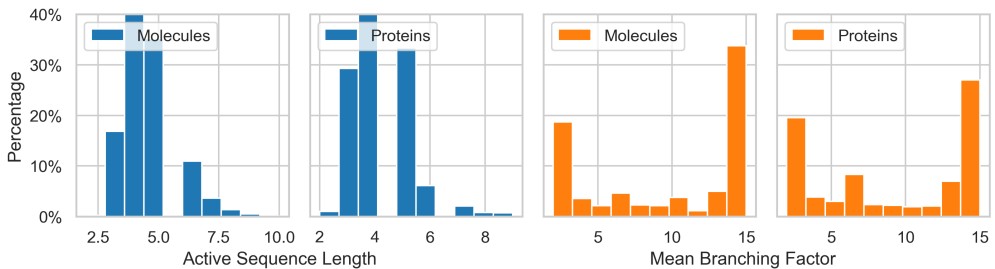

Figure 4: Constrained generation space structure analysis: Distribution of active sequence lengths for molecules and proteins (left), and mean branching factors for molecules and proteins (right). Active sequence length represents the number of positions requiring actual prediction decisions, while branching factor indicates the number of valid token choices at each active position.

Each figure below shows the distribution of active sequence lengths (number of positions requiring actual prediction decisions) and branching factors (number of valid token choices at each active position)

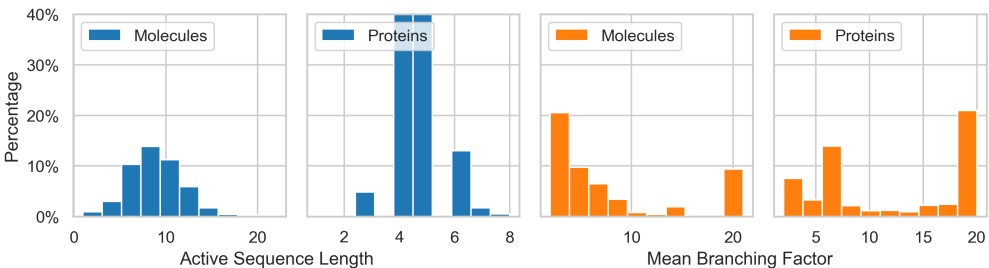

Figure 5: Active sequence lengths and branching factors for BioSNAP dataset with raw representations.

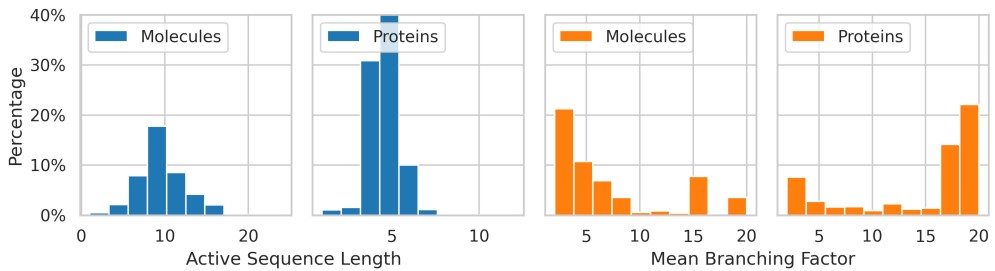

Figure 6: Active sequence lengths and branching factors for DrugBank dataset with raw representations.

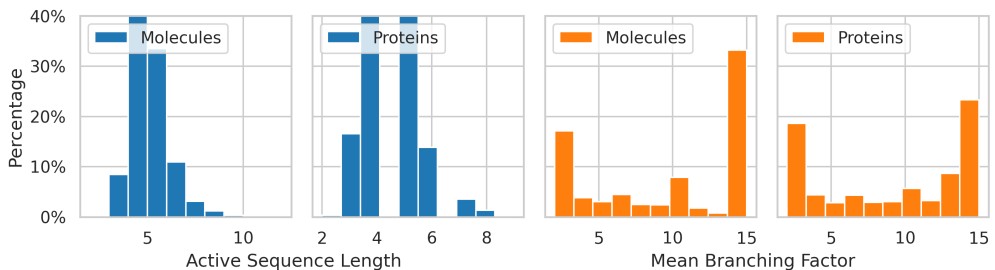

Figure 7: Active sequence lengths and branching factors for DrugBank dataset with quantized representations.

## B Hyperparameter Selection

Model hyperparameters were selected using a greedy search approach. This process involved iteratively tuning individual hyperparameters based on their performance on the validation set, aiming to maximize the validation AUC metric. The chosen values for the main model, meta-model, and training process are detailed below.

### B.1 Main Model Hyperparameters

Model hyperparameters were selected using a greedy search approach. This process involved iteratively tuning individual hyperparameters based on their performance on the validation set, aiming to maximize the validation AUC metric. The chosen values for the main model, meta-model, and training process are detailed below. In addition to the parameters listed in the tables, the main Transformer model utilized a hidden dimension of 512. The encoder outputs were processed using global average pooling. The I-RVQ module employed the standard k-means algorithm from scikit-learn [15] with 15 clusters. Both the main model and the meta-model were trained using the AdamW optimizer. The main model training ran for up to 25,000 steps, with early stopping based on performance on the validation AUC. The meta-model was trained similarly, with its own validation criteria (details in Table 8). Training was performed on an L40 GPU with 64GB of memory. The main model training required approximately 6 hours, and the meta-model training took around 5 minutes.

Table 6: Main Model Hyperparameters

| Parameter | Selected Value | Search |
|---|---|---|
| Number of Layers (Decoder) | 8 | 2,4,6,8,12 |
| Number of Attention Heads (Decoder) | 8 | 2,4,8,16 |
| Feedforward Dimension (Decoder) | 2048 | 4 * Hidden Size |
| Dropout | 0.2 | 0,0.1,0.2,0.3,0.5 |
| Bottleneck Dimension | 128 | -1,64,128,256,512 |
| Pooling | True | True,False |

## B.2 Main Model Training Hyperparameters

The training process for the main generative model used these hyperparameters, selected based on validation performance:

Table 7: Main Model Training Hyperparameters

| Parameter | Selected Value | Search/Notes |
|---|---|---|
| Learning Rate | 1e-4 | 1e-3,1e-4,1e-5 |
| LR Scheduler | Constant | Constant,Linear |
| Batch Size | 64 | 16,64,256,1024 |

## B.3 Meta-Model Hyperparameters

The discriminative meta-model, used for calibrating generation statistics into interaction scores, was configured with the following hyperparameters:

Table 8: Meta-Model Hyperparameters

| Parameter | Selected Value | Search/Notes |
|---|---|---|
| Number of Layers | 1 | 1,2,3,4,5 |
| Hidden Dimensions | 128 | 32,64,128,256 |
| Number of Attention Heads | 4 | 2,4,8 |
| Learning Rate | 1e-4 | 1e-3,1e-4,1e-5 |
| LR Scheduler | Constant | Constant,Linear |
| Batch Size | 64 | 16,64,256,1024 |

