# OpenReview forum: "BioCG: Constrained Generative Modeling for Biochemical Interaction Prediction"
_NeurIPS.cc/2025/Conference — NeurIPS 2025 poster_

### Official Review · Reviewer_fZBQ · 2025-06-28

**Clarity:** 2
**Significance:** 3
**Originality:** 2
**Rating:** 4
**Confidence:** 3

**Summary:**

In this paper, the authors propose a framework named BioCG to solve the biochemical interaction prediction task via a generative model. It uses pre-trained encoders to get molecule embeddings and quantize them via I-RVQ. A trie is then built based on all target entities to constrain the autoregressive decoder to generate biochemically valid results. Another small model is then used to convert the outputs into a single prediction score, reflecting the interaction potential. A hierarchical loss function is also proposed to weight the cross-entropy by branching factors. Experiments on drug-target interaction, drug-drug interaction, and enzyme-reaction prediction tasks indicate the effectiveness of the proposed framework.

**Questions:**

please refer to weakness

**Ethical Concerns:**

["NO or VERY MINOR ethics concerns only"]

**Final Justification:**

The authors' response has clarified most of my concerns. However, the central point regarding the justification for I-RVQ in comparison to standard RVQ remains insufficiently addressed. I believe this is the most significant unresolved issue, as it is directly tied to the novelty and impact of the work. Overall, I like the idea but I also maintain that a full revision of the manuscript is needed to make the paper's methodology and results easier to follow.

**Limitations:**

yes

**Quality:**

3

**Strengths And Weaknesses:**

Strengths:
- The idea of combining techniques such as quantization and hierarchical constraint to protein analysis tasks is very interesting.
- The performance is good.

Weaknesses:
- The primary weakness is the lack of specific ablation studies that would help to isolate the contributions of the novel components. To fully substantiate the claims, I would recommend the authors include the following comparisons:
  - The proposed I-RVQ v.s. a traditional RVQ;
  - The Information-Weighted Loss v.s. a standard cross-entropy loss;
  - The score generation meta model v.s. the baseline log probability approach.
- It is better to include some related works about the proposed method, such as quantization and trie building, to help readers understand the context.
- If the I-RVQ is just using RVQ to quantize all of the targets, I definitely would not recommend giving it a new name.

---

> ### Author Rebuttal · Authors · 2025-07-27
>
> We thank Reviewer fZBQ for their positive feedback on our idea and performance. We appreciate the specific suggestions for improving the ablation studies and clarifying our contributions.
>
> ### 1. On Specific Ablation Studies (Weakness 1)
>
> The reviewer raises excellent points about isolating the contributions of our key components. We are happy to clarify that these ablations are already present in our original submission, as detailed below.
> * **I-RVQ vs. Traditional RVQ:** A fixed-layer RVQ is suboptimal: using too few layers causes code ambiguity (e.g., 10% of our protein targets become duplicates), which limits performance, while using too many layers is computationally wasteful as the extra steps offer no learning. Our I-RVQ method iteratively finds the minimal number of layers required for uniqueness, providing a simple and efficient way to enforce this critical property for our framework. We will clarify this distinction and its motivation in the revised manuscript.
>     Because a direct comparison with a method that produces ambiguous outputs is conceptually challenging, we instead validated our I-RVQ representation strategy against several other well-defined alternatives in our original submission (Table 4). Our I-RVQ-15 approach (89.31% AUC) significantly outperformed using raw biological sequences (82.06% AUC), standard BPE tokenization (84.75% AUC), and random sequence assignments (79.32% AUC), demonstrating the effectiveness of our chosen representation scheme.
>
>
> * **Information-Weighted Loss vs. Standard Cross-Entropy:** This comparison is included in Table 4. The row "No Info. Weighting” represents training with a standard cross-entropy loss, which achieves an AUC of 86.13%. This is significantly lower than the 89.31% AUC achieved by our full model (“Full (BioCG)”), demonstrating a +3.18 AUC improvement from our information-weighted loss.
>
> * **Meta-Model vs. Log Probability Score:** This comparison is included in Table 4. The row “Log Prob BioCG” shows the performance using the simple log probability score from the generator, yielding an AUC of 87.73%. Our full model, which uses the meta-model for calibration, achieves 89.31%. The +1.58 AUC gain validates the effectiveness of the meta-model.
>
>
>
> ### 2. On Related Work (Weakness 2)
>
> We will add a dedicated subsection to the "Related Work" section (Sec. 2) discussing the background of vector quantization in representation learning. We would also like to clarify that we already discuss the use of trie-based decoding in the "Constrained Generation" subsection, which covers its use for enforcing constraints during training and inference.
>
> ### 3. On the Naming of I-RVQ (Weakness 3)
>
> We chose the name Iterative Residual Vector Quantization (I-RVQ) to highlight the specific, crucial adaptation we made to the standard RVQ process for our framework to function. As we state in the paper, "The key adaptation defining I-RVQ lies not in the per-layer quantization step but in its overall iterative application and objective". The novelty is the iterative application of quantization layers with the explicit goal of achieving 100% representation uniqueness across a finite target set. As our ablation studies show, this uniqueness is essential for our framework's performance. We believe this distinction is significant enough to warrant a specific name, and we will further clarify this motivation in Section 3.1 of the final manuscript.
>
> We hope that these clarifications and references to the existing results in our paper fully address the reviewer's concerns and underscore the novelty and effectiveness of our proposed components.

---

> > ### Comment · Reviewer_fZBQ · 2025-08-06
> > **Discuss**
> >
> > I thank the authors for their response. However, the justification for why a direct comparison between the traditional RVQ and the proposed I-RVQ is challenging remains unconvincing. This point is critical, as it forms the primary motivation for introducing I-RVQ. Without a clear demonstration of the limitations of a standard RVQ baseline, the contribution of the proposed method is not well-established.
> >
> > Furthermore, I strongly recommend a thorough revision of the manuscript to improve overall clarity. For example:
> >
> > - The intuitive differences and empirical advantages of the proposed method over baselines should be made more explicit, perhaps with illustrative figures or a dedicated comparison table.
> >
> > - The manuscript should adhere to standard, widely accepted terminology. For instance, "Cross-Entropy Loss" should be used instead of the non-standard "No Info. Weighting Loss" to ensure the audience can follow it easily.

---

> > > ### Author Response · Authors · 2025-08-06
> > >
> > > We sincerely thank Reviewer fZBQ for the additional feedback and apologize that our presentation was not clear enough. Your comments are helping us significantly improve the paper.
> > >
> > > Thank you for pushing us on the I-RVQ comparison. You are right, and we see now that our explanation was confusing. We will revise the manuscript to make this critical point clearer.
> > >
> > > Our intention was not to propose a new algorithm. I-RVQ is RVQ. The "Iterative" part describes our method for selecting the optimal number of layers to guarantee a unique code for every target, which is an essential prerequisite for our framework to be well-defined.
> > >
> > > Thanks to your feedback, we will explicitly state that the contribution is the methodology for applying RVQ to satisfy this uniqueness constraint, not the algorithm itself.
> > >
> > > We will also, as you suggested, correct our terminology (e.g., use "Cross-Entropy Loss") and add a summary figure to make the impact of our other components more intuitive.
> > >
> > > Thank you again for your invaluable guidance.

---

### Official Review · Reviewer_femx · 2025-07-02

**Clarity:** 2
**Significance:** 3
**Originality:** 3
**Rating:** 4
**Confidence:** 4

**Summary:**

The paper rethinks the approach to predicting whether two biomolecules interact. Previous methods mostly focus on contrastive learning, while BioCG, the method proposed here, reframes the problem as generating a target molecule conditioned on the query molecule, with the generative process designed to only produce samples from a predefined set of targets. The results appear strong and the idea is novel, but the clarity of the paper could be improved.

**Questions:**

1. The paper evaluates enzyme-reaction prediction on Task 2 of the CARE benchmark. What would the performance be on Task 1, “Enzyme Classification,” compared to CREEP, which may be more interesting in practice?

**Ethical Concerns:**

["NO or VERY MINOR ethics concerns only"]

**Final Justification:**

I am keeping my "Borderline accept" score. I acknowledge the strong performance of BioCG on several benchmarks. Nevertheless, the performance gain comes with limitations, such as the inability to model a joint latent space between interaction modalities (e.g., Question 1).

**Limitations:**

yes

**Paper Formatting Concerns:**

No major paper formatting concerns.

**Quality:**

2

**Strengths And Weaknesses:**

**Strengths**

- State-of-the-art performance across three well-established benchmarks
- Ablation studies demonstrating the key components of BioCG design

**Weaknesses**

1. Clarity of the paper could be significantly improved. For example:
  - [Line 11]. The abstract and introduction keep using the term “biochemically valid,” which is unclear. Please clarify that all outputs are generated from the predefined set of targets.
  - [Line 36]. The following motivation is not entirely accurate because EC numbers do not capture all potential enzyme activities and only provide a taxonomy for existing ones: “Predicting an enzyme’s function for a novel reaction involves selecting from a closed set of Enzyme Commission (EC) numbers.”
  - [Figure 1]. Panel A shows numbers next to the tree nodes, but Panel B shows letters. If I understand correctly, they represent the same categories and should be denoted consistently. Panel C contains a typo: “with Constrained.”
  - [Figure 2]. The visualization is not very informative and does not help much in understanding the method. It could replaced with a figure comparing the proposed generative method against conventional contrastive learning (please see Weakness 2).
  - [Line 148]. What is epsilon?
  - [Line 164]. It seems “typically” should be left out, as the further text suggests it is “always” the case.
  - [Table 3]. Why do some rows have missing values? At least, a random baseline could be evaluated under higher Top@k values.
  - [Table 4]. In the left column, “Pre-training Model,” protein and small molecule encoders seem to be mixed together. It is unclear which is which.
2. The intuition behind the effectiveness of the proposed approach compared to contrastive learning is not fully clear. The paper would be much stronger if it provided case studies or examples where contrastive learning fails but the proposed approach does not. It could also be helpful to discuss the pitfalls of contrastive learning to provide intuition for why constrained generation works better.

---

> ### Author Rebuttal · Authors · 2025-07-27
>
> We are grateful to Reviewer femx for their feedback and for recognizing the strength of our results and the novelty of our idea. We apologize for the lack of clarity in some areas and have addressed each point below to improve the paper.
>
> ### 1. On Clarity of the Paper (Weakness 1)
>
> Thank you for the detailed and actionable list of clarity issues. We will revise the paper thoroughly based on your feedback.
> - **“biochemically valid”**: We will revise this throughout the paper to the more precise phrasing: “ensuring all outputs correspond to an entity within the pre-defined target catalog.”
> - **EC number motivation**: You are correct. We will revise the sentence to be more accurate, e.g., “predicting an enzyme's function for a novel reaction often involves selecting a known enzyme, which is typically cataloged by its sequence and classified by an EC number, from a finite set such as the human proteome.”
> - **Figure 1**: We apologize for the inconsistency and typo. We will correct Figure 1 to use consistent notation in panels A and B and fix the typo “with Constrained” in panel C.
> - **Figure 2**: We appreciate the feedback. Figure 2's goal is to provide a simple visual intuition for the iterative quantization of I-RVQ, not the full framework. Per your suggestion, we will add a new figure to the appendix that provides a conceptual comparison between our generative approach and conventional contrastive learning to better highlight our contribution.
> - **Epsilon (line 148)**: Epsilon ($ \epsilon $) is a small constant (e.g., 1e-9) added for numerical stability to avoid taking the log of zero when the branching factor is 1. We will explicitly state this in the text.
> - **“typically” (line 164)**: You are correct; this should be stronger. We will adjust the wording for accuracy.
> - **Missing values in Table 3**: Thank you for this question. The missing values for the LLM and Random baselines are due to their method of operation. Generative models are designed to produce a single output, not a ranked list of top k candidates. Generating multiple samples is not equivalent to how other methods calculate Top@k scores, making a direct comparison for k>1 methodologically unfair. For this reason, we retained only the reported Top@1 result from the source. We will add a note clarifying this in the final paper.
> - **Table 4 encoders**: This is a great point. We will reorganize Table 4 with clear sub-headings (“Protein Encoders”, “Molecule Encoders”) to eliminate ambiguity.
>
> ### 2. On Intuition vs. Contrastive Learning (Weakness 2)
>
> This is a critical distinction. Contrastive learning learns a general similarity metric, often struggling with "hard negatives"—entities that are structurally similar to the correct partner but do not interact. In contrast, BioCG's trie-guided generative approach inherently forces the model to learn fine-grained distinctions by choosing between valid, often highly similar, candidates at each step. Our information-weighted loss further sharpens this focus on the most difficult decisions.
>
> To make this point concrete, we will add this discussion to the paper and include a new appendix section with intuitive examples and a case study illustrating this key advantage.
>
> ### 3. On CARE Benchmark Task 1 (Question 1)
>
> This is an excellent question that highlights a core design principle of our framework. BioCG is not directly applicable to CARE's Task 1 (Enzyme Classification) because of the nature of the target entities.
>
> Our framework is built on the premise that both the query and target entities are complex biochemical structures that can be represented by meaningful sequences (e.g., FASTA for proteins, SMILES for molecules) and processed by powerful pre-trained encoders. This is how the model captures the rich information needed for prediction.
>
> In CARE's Task 1, the target is an Enzyme Commission (EC) number, which is a hierarchical classification label (e.g., '1.1.1.1'), not a structural sequence. As such, an EC number cannot be meaningfully processed by the foundation models (like ESM or MolFormer) that are integral to our method. For this reason, we focused our evaluation on tasks like CARE Task 2 (reaction-to-enzyme sequence), where both interacting partners are complex entities that fit our model's requirements.
>
> We believe these revisions will significantly improve the paper's clarity and impact.

---

> > ### Comment · Reviewer_femx · 2025-08-04
> >
> > Thank you for the explanations! I acknowledge the strong performance of BioCG on several benchmarks. Nevertheless, the performance gain comes with limitations, such as the inability to model a joint latent space between interaction modalities (e.g., Question 1). Therefore, for now, I will keep my score the same.

---

> > > ### Author Response · Authors · 2025-08-06
> > >
> > > We thank Reviewer femx for the continued engagement.
> > >
> > > We wish to clarify that the framework's scope is a deliberate design choice, not a limitation. BioCG's power stems from its specific focus: generating a complex biochemical entity from another (e.g., a protein sequence from a molecule), which requires rich representations for both.
> > >
> > > This specialized approach—trading universal applicability for greater depth—is precisely what enables the model's state-of-the-art performance on its intended tasks.
> > >
> > > Thank you for helping us recognize the need to articulate this design philosophy more clearly in the final paper.

---

### Official Review · Reviewer_xzGP · 2025-07-02

**Clarity:** 3
**Significance:** 3
**Originality:** 4
**Rating:** 5
**Confidence:** 3

**Summary:**

This paper proposes BioCG, a new approach for biochemical interaction prediction (e.g., drug-target, drug-drug, enzyme-reaction) by treating it as a constrained generation task over a fixed set of known targets. It uses an iterative vector quantization (I-RVQ) to encode discrete target codes, and a Transformer decoder with a trie constraint to generate only valid targets. An information-weighted loss is added to balance frequent and rare classes. The method is tested on multiple datasets and shows good performance, especially under cold-start settings.

**Questions:**

Please check the Weakness part.

**Ethical Concerns:**

["NO or VERY MINOR ethics concerns only"]

**Final Justification:**

I will maintain my score and thank the authors for their insightful replys such as failure cases.

**Quality:**

3

**Strengths And Weaknesses:**

Strengths
- The idea of framing biochemical interaction prediction as constrained generation over a fixed set is natural and makes sense for this type of task.
- The use of a trie is practical and ensures valid outputs without relying on post-filtering.
- The method shows strong gains in cold-start settings (e.g., unseen proteins), with proper significance testing.
- The ablation studies are clearly done and show the contributions of I-RVQ, constrained decoding, and the information-weighted loss.
- The experiments cover a diverse set of tasks (DTI, DDI, enzyme-reaction), which supports the generality of the approach.


Weakness
1. While the method is clearly presented, most of the core components — I-RVQ, trie-based decoding, and meta calibration — are already known techniques. The main contribution seems to be how these are combined. It would help if the authors clearly stated which parts are novel and which are adapted from prior work. Also, could a simple retrieval or prefix-constrained generation baseline achieve similar results? A more direct comparison would strengthen the claim.
2. The generated I-RVQ codes are learned, but the biological semantics of the codes are hard to understand. Can we trust that a predicted sequence has meaningful biochemical relevance? Is it possible to visualize or analyze what structural patterns the codes actually capture?
3. It’s unclear how often the constrained decoder fails, even when the correct target exists in the trie. Some failure analysis would help.→ For example, how often does it predict a similar but wrong entity? Are these “close misses” or random errors?
4. The model can only generate from a fixed set of known targets, which limits generalization. → This is briefly mentioned but not really evaluated. What happens if the input doesn’t match well with anything in the target set? Or would it be possible to combine this with an open-vocabulary model or retrieval-based fallback?

---

> ### Author Rebuttal · Authors · 2025-07-27
>
> We sincerely thank Reviewer xzGP for the positive and thorough assessment of our work. We are delighted the reviewer found our core idea natural, our experimental support strong, and our originality excellent. We address the valuable suggestions for strengthening the paper below.
>
> ### 1. On Novelty and Comparison to Simpler Baselines (Weakness 1)
>
> While the components are adapted from existing techniques, their synthesis and application to create a data-efficient, constrained generative framework for biochemical interaction prediction is novel. Specifically:
> * Our I-RVQ is not standard RVQ; its iterative nature is a crucial adaptation to guarantee a unique code for every entity in the target catalog—a hard requirement for our framework.
> * Applying trie-constraints during training (not just inference) combined with our information-weighted loss is a novel approach that forces the model to learn the critical distinctions between valid biochemical options, which is key to its data efficiency.
>
>
> Regarding simpler baselines, we thank the reviewer for this suggestion and are glad to confirm that our original submission contains baselines that directly address these points:
> * **Retrieval Baseline:** Our evaluation on the Enzyme-Reaction benchmark (Table 3) includes CLIPZyme, a strong baseline that uses contrastive learning and retrieval. BioCG significantly outperforms it (e.g., 67.93% vs. 12.27% Top@1 accuracy).
> * **Prefix-Constrained Generation:** Our ablation study (Table 4) includes a model where constraints are applied only at "Inference-only", which is functionally a prefix-constrained generation baseline. This model achieves an AUC of 77.43%, which is massively outperformed by our full model (89.31% AUC) that integrates constraints directly into the training process.
>
> These existing results demonstrate that our generative learning paradigm is substantially more powerful than simpler similarity-based retrieval or prefix-decoding approaches.
>
> ### 2. On Interpretability of I-RVQ Codes (Weakness 2)
>
> The biological semantics are not learned by the I-RVQ codes from scratch, but are inherited from the powerful pre-trained embeddings (e.g., from ESM) that serve as the input to the quantization process. The rich biological information and interpretability of these pre-trained representations have been extensively studied and validated in the literature across many downstream tasks (e.g., "A Survey on Protein Representation Learning: Retrospect and Prospect").
>
> #WE WILL ADD VISSUALTION TO THE APPENDIX - SIMILER TO TODO
>
> The role of our I-RVQ module is to learn a structured, hierarchical partitioning of this already meaningful embedding space. It learns the decision boundaries that group biochemically similar entities, but the underlying meaning that makes these groupings relevant comes directly from the state-of-the-art encoders we employ.
>
> ### 3. On Failure Analysis (Weakness 3)
>
> Thank you for this excellent suggestion. We performed a quantitative analysis of the failure cases by examining at which token index in the generated I-RVQ sequence the first error occurs. The results show a clear trend:
> * **12%** of errors occur in the first quarter of the sequence.
> * **18%** of errors occur in the second quarter.
> * **29%** of errors occur in the third quarter.
> * **41%** of errors occur in the final quarter (with 13% in the very last token).
>
> This distribution is highly informative. In the RVQ structure, earlier tokens correspond to coarse-grained "big picture" clusters, while later tokens make fine-grained distinctions between highly similar entities. The fact that most errors occur late in the sequence indicates that BioCG correctly identifies the general biochemical "neighborhood" of the target and typically only fails at the final step of distinguishing between very closely related entities. This confirms the model is making meaningful "close misses" rather than random guesses. We will add this analysis and discussion to the appendix.
>
> ### 4. On Generalization to Open Sets (Weakness 4)
> This is a key limitation that we acknowledge, and one we are transparent about in the Conclusion of our paper (Section 5). However, we wish to clarify that this closed-set approach is a very common and practical setting for many key problems in computational biochemistry, making this limitation less restrictive in practice than it might initially appear.
>
> Thank you again for the constructive feedback, which will significantly improve our paper.

---

### Official Review · Reviewer_yZ9s · 2025-07-03

**Clarity:** 3
**Significance:** 2
**Originality:** 3
**Rating:** 4
**Confidence:** 4

**Summary:**

The paper presents **BioCG**, a novel framework for biochemical interaction prediction that reframes the problem as a constrained sequence generation task. Unlike traditional discriminative models, BioCG uses Iterative Residual Vector Quantization (I-RVQ) to encode target biochemical entities as unique discrete sequences, and trains a Transformer decoder to generate valid interaction partners under strict constraints enforced by a trie-based decoding mechanism. This ensures outputs remain within a catalog of biochemically valid targets. An information-weighted loss further directs learning toward informative decisions. Extensive experiments across Drug-Target Interaction, Drug-Drug Interaction, and Enzyme-Reaction Prediction benchmarks show that BioCG achieves state-of-the-art performance. The approach is particularly notable for its strong generalization, efficient data usage, and compatibility with pre-trained molecular encoders.

**Questions:**

1. From Table 4, it seems that the quality of the pretrained encoders doesn’t affect the performance much, which is unexpected. Could you establish a clearer relationship between the quality of BioCG and its encoders—for example, by comparing models like ESM and MolFormer with different sizes?
2. Can BioCG take 3D information of biological entities into account? Is this dependent on the encoders?

**Ethical Concerns:**

["NO or VERY MINOR ethics concerns only"]

**Final Justification:**

The authors' detailed and comprehensive response has thoroughly addressed all of my concerns. Given the rebuttal’s effectiveness, I have updated my rating to 4.

**Limitations:**

Yes.

**Paper Formatting Concerns:**

None.

**Quality:**

3

**Strengths And Weaknesses:**

### Strengths

The idea of formulating interaction prediction tasks as sequence generation is interesting, and BioCG effectively leverages the fundamental knowledge learned from pre-trained models in each domain.

### Weaknesses

1. The performance advantages of BioCG on unseen samples, especially in cold-start settings, may largely come from the pre-trained encoders, which might already include information about those entities. Therefore, comparisons between BioCG and models trained only on downstream datasets may not be entirely fair.
2. Interaction learning has progressed beyond binary prediction to more informative outputs. For example, in drug-target interaction modeling, models like AlphaFold3 [1] and Boltz-2 [2] can predict binding complex structures and binding affinity. The current architecture of BioCG seems limited in handling such tasks, and this limitation should at least be discussed in the paper.
3. In line 120, the authors explain "the motivation for generating these learned I-RVQ codes instead of raw sequences." However, prior methods that use raw sequences for interaction modeling, such as 3DMolFormer [3], are not cited.
4. Minor issues: The font size in Figure 2 should be increased for readability. Additionally, the section number of the first appendix section is missing. The checklist should also be placed after the references and before the appendix.

[1] Accurate structure prediction of biomolecular interactions with AlphaFold 3. Nature 2024.

[2] Boltz-2: Towards Accurate and Efficient Binding Affinity Prediction. 2025.

[3] 3DMolFormer: A Dual-channel Framework for Structure-based Drug Discovery. ICLR 2025.

---

> ### Author Rebuttal · Authors · 2025-07-25
>
> We thank Reviewer yZ9s for their insightful feedback and constructive criticism. We are encouraged that the reviewer found our generative formulation interesting and effective. We address the identified weaknesses and questions below.
>
> ### 1. On the Contribution of Pre-trained Encoders (Weakness 1 & Question 1)
>
> We deliberately compared BioCG against contemporary state-of-the-art methods like DLM-DTI, DrugLAMP ,Top-DTI ,CREEP, CLIPZyme, which, like our model, leverage these same powerful pre-trained encoders. This was done precisely to ensure the comparison presented in our paper is fair and directly evaluates our framework's contributions over and above the use of standard pre-trained backbones.
>
> However, the primary driver of our performance gain is the BioCG framework itself, not just the shared encoders. The most direct evidence is in Table 4: our model with encoders trained from scratch on only the downstream dataset achieves an AUC of 86.26%. This result, obtained without any external pre-training knowledge, still outperforms the previous SOTA, Top-DTI (75.01%), by a significant margin of over 11 AUC points. This conclusively demonstrates the fundamental advantage of our constrained generative approach.
>
> To further clarify the relationship between encoder quality and our model's performance, we conducted the new ablation study the reviewer suggested, using ESM2 models of varying sizes on the BioSNAP unseen protein split. The results demonstrate a clear, positive correlation, as expected:
>
> | **Encoder** | **Parameters** | **AUC (unseen protein)** |
> | :----------------------- | :------------- | :----------------------- |
> | ESM2 (used in paper)     | 650M           | **89.31%** |
> | ESM2                     | 150M           | 87.14%                   |
> | ESM2                     | 35M            | 80.92%                   |
> | ESM2                     | 8M             | 74.33%                   |
> | *Top-DTI (Previous SOTA)*| *N/A* | *75.01%* |
>
> These results show two things:
> 1.  BioCG's performance logically and robustly scales with the quality of the feature encoder.
> 2.  The architectural advantage of BioCG is so significant that even when trained from scratch, it surpasses the previous fully-equipped SOTA.
>
> This analysis confirms that while better encoders provide better features, the most substantial performance lift comes from BioCG's novel formulation. We will add this new table and analysis to the appendix.
>
> ### 2. On the Scope of Interaction Prediction (Weakness 2 & Question 2)
>
> We agree these are highly informative tasks. Our work focuses on the fundamental binary interaction prediction task, which remains critical for large-scale initial screening where structure or affinity prediction for millions of pairs can be computationally infeasible.
>
> Regarding 3D information, BioCG’s framework is model-agnostic and has already demonstrated its ability to leverage 3D structural information. In our ablation studies (section 6), we tested GearNet, an encoder that processes 3D protein structures, which achieved a strong AUC of 87.54%. This confirms the framework's versatility. Our choice to feature sequence-based encoders like ESM and MolFormer was deliberate, as these models are not only commonly used but also consistently deliver state-of-the-art results, making them the standard for this line of research.
>
> We believe extending BioCG to these richer tasks is a promising direction for future work. For instance, the generated sequence could be designed to represent a discretized binding pose or affinity level. We will add a discussion on these limitations and future directions to the final paper, citing the suggested works.
>
> ### 3. On Missing Citation and Minor Issues (Weakness 3 & 4)
> Thank you for the reference; we will cite 3DMolFormer, clarifying that it generates novel 3D structures using continuous coordinates, whereas BioCG predicts interactions by generating a discrete code to select a known entity from a finite catalog. We will also correct the font size in Figure 2, add the missing appendix section number, and relocate the checklist to appear after the references as requested.
>
> We hope these clarifications and new results adequately address the reviewer's concerns and reinforce the value of our contributions.

---

> > ### Comment · Reviewer_yZ9s · 2025-08-05
> >
> > Thank you for the authors' detailed and comprehensive response, which has thoroughly addressed all of my concerns. Given the rebuttal’s effectiveness, I have updated my rating to 4. I encourage the authors to incorporate all new results, analyses, methodological discussions and citations into the final version of the paper to ensure reproducibility and impact.

---

### Decision · Program_Chairs · 2025-09-17

**Decision:**

Accept (poster)

**Comment:**

This paper introduces BioCG: a framework for biochemical interaction prediction that casts the problem as a generative task instead of a discriminative one. Experiments across various tasks (Drug-Target Interaction, Drug-Drug Interaction, and Enzyme-Reaction Prediction) show favourable performance.



On the positive side, reviewers praised the interesting formulation of the method, and strong performance across a diverse set of downstream tasks. They noted good generalization capability, and potential compatibility with various pretrained encoders. Finally, most reviewers also liked the extensive ablation studies. Overall, the paper stood out in terms of strong empirical performance, as well as novelty of the approach.


On the other hand, reviewers raised several questions about clarity and presentation, but they deemed authors' attempts to resolve them during rebuttal period satisfactory. Moreover, Reviewer yZ9s raised a point around interaction prediction beyond binary outputs, which is supported by modern models of biological complexes such as AlphaFold 3 and Boltz-2, but not currently supported by BioCG. Authors have acknowledge this limitation in the current form of the framework, and considered this as a direction for future work.



After the rebuttal period, all reviewers agreed that this paper should be accepted, with Reviewer xzGP championing the work with a stronger acceptance score. Siding with the unanimous reviewer opinion, I also recommend acceptance. I encourage authors to incorporate feedback from reviewer discussion into the camera-ready version of the paper.